# A Biomechanical Comparison of the Safety-Bar, High-Bar and Low-Bar Squat around the Sticking Region among Recreationally Resistance-Trained Men and Women

**DOI:** 10.3390/ijerph18168351

**Published:** 2021-08-06

**Authors:** Eirik Kristiansen, Stian Larsen, Markus E. Haugen, Eric Helms, Roland van den Tillaar

**Affiliations:** 1Department of Sport Sciences and Physical Education, Nord University, 7600 Levanger, Norway; ek1105@hotmail.com (E.K.); stianandrelarsen@live.no (S.L.); Haugen@helseogprestasjon.no (M.E.H.); 2Sports Performance Research Institute New Zealand, Auckland University of Technology, Auckland 1142, New Zealand; eric.helms@aut.ac.nz

**Keywords:** squat, strength, inverse dynamics, myoelectric activity

## Abstract

Barbell placement can affect squat performance around the sticking region. This study compared kinematics, kinetics, and myoelectric activity of the safety-bar squat with the high-bar, and low-bar squat around the sticking region. Six recreationally resistance-trained men (26.3 ± 3.1 years, body mass: 81 ± 7.7 kg) and eight women (22.1 ± 2.2 years, body mass: 65.7 ± 10.5 kg) performed three repetition maximums in all three squat conditions. The participants lifted the least load with the safety bar followed by the high-bar and then the low-bar squat. Greater myoelectric activity of the gluteus maximus was observed during safety-bar squats than high-bar squats. Also, larger knee extension moments were observed for the safety bar compared with low-bar squat. Men had higher myoelectric activity in the safety-bar condition for the gluteus maximus during all regions in comparison with women, and greater knee valgus at the second occurrence of peak barbell velocity. Our findings suggest that the more upright torso inclination during the safety-bar could allow greater gluteus maximus contribution to the hip extensor moment. Moreover, low-bar squats allowed the greatest loads to be lifted, followed by the high-bar and safety-bar squats, possibly due to the larger hip moments and similar knee moments compared to the other squat conditions. Therefore, when the goal is to lift the greatest load possible among recreationally trained men and women, they should first attempt squatting with a low-bar technique, and if the goal is to increase myoelectric activity in the gluteus maximus, a safety-bar squat may be the more effective than the high- bar squat.

## 1. Introduction

The back squat is a commonly used exercise by recreational lifters and athletes, both in training and competition [1]. A back squat is performed by placing the bar on the back, either on top of the trapezius muscle or below the top of the trapezius muscle (high-bar and low-bar conditions, respectively). Only in recent years has squatting with a “safety-bar” (a cambered barbell with handles) become popular. Hence, only one study has investigated the safety-bar squats, finding that while it requires less shoulder mobility and puts less force on the shoulder and elbow joints, three-repetition maximum (3-RM) squats are 11.3% lower compared to squatting with a traditional barbell [2]. Thus, the safety-bar could be seen as ineffective for maximizing strength gains, while being a useful alternative exercise since it places less stress on the lower back due to lesser torso inclination. However, Hecker, Carlson and Lawrence [2] did not report if the Olympic back squat exercise was with a high-bar or a low-bar barbell placement. As stated, they found that squatting with a safety-bar produced a more upright torso, resulting in less hip flexion and a slight decrease in dorsiflexion. Furthermore, they found less myoelectric activity in the gluteus maximus, vastus medialis, rectus femoris, erector spinae, latissimus dorsi, middle trapezius, and the external obliques when squatting with a safety-bar compared to a traditional bar. The lower 3-RM during safety-bar squats could perhaps be explained in the shift of center of mass. The forward camber of the safety-bar position and the shift of center of mass could potentially make the exercise more difficult since it places greater demands to balance the center of mass for proper force production [2].

When performing the barbell back squat with high-bar or low-bar barbell placement, studies often analyze lower-body kinematics [3]. According to Glassbrook, Helms, Brown and Storey [3], the main differences between high-bar and low-bar squats are a greater forward torso lean during the low-bar squat. With this forward lean, the force increases on the hip joint compared to the knee joint. This leads to a greater internal moment arm for the gluteus maximus and, therefore, increases the potential for lifting heavier loads with a low-bar position compared to the high-bar position [3,4]. For the knee joint, 11 of 14 studies showed greater knee flexion with the high-bar squat, resulting in a deeper bottom position [3]. Glassbrook, Helms, Brown and Storey [3] also reported differences between the high-bar and low-bar conditions in myoelectric activity, where forward lean in the low-bar squat often results in greater myoelectric activity in the posterior chain, while the greater knee flexion angles together with a more upright torso for the high-bar squat often result in greater quadriceps myoelectric activity.

When squatting with loads over 85% of 1-RM, a sticking region occurs during ascent of the squat [5,6]. The sticking region is where lifters have a higher chance of failing a squat [7,8]. The sticking region is one of the three regions of the ascent phase, which starts with a pre-sticking region (lowest barbell height: v_0_) to the first located peak velocity (v_max1_). The sticking region start from this point to the first located minimum velocity (v_min_) in the ascent phase, right before the barbell accelerates again to the next peak velocity of the barbell (v_max2_), which is the post sticking region. It is suggested that is there a co-contraction between the quadriceps and the gluteus muscles, which causes a deceleration in the ascent and thereby the sticking region in the back squat [7]. This may occur due to less favorable joint angles and longer moment arms of the muscles involved. Furthermore, van den Tillaar, et al. [8] investigated the effects of high-bar and low-bar squats on joint kinematics and myoelectric activity around the sticking region. The authors found no significant differences in barbell and joint kinematics around the sticking region between techniques when absolute load, depth, and stance width were controlled. However, they observed an increase in myoelectric activity in the rectus femoris, vastus medialis, and the lower part of the erector spinae with the high-bar position compared to the low-bar position.

To the authors’ best knowledge, safety-bar squats have not previously been compared to high-bar and low-bar squats. Therefore, the purpose of this study was to compare barbell and joint kinematics, kinetics, and myoelectric activity of the safety-bar squat with the high-bar and low-bar squat around the sticking region. We hypothesized that squatting with a safety-bar or high-bar would produce a more vertical torso position due to less peak hip flexion and a forward shift in center of mass, creating greater knee joint moments and thereby, greater myoelectric activity during these squat conditions for the knee extensor muscles. We further hypothesized that the low-bar squat would result in more hip flexion, creating greater hip joint moments and hip extensor myoelectric activity compared to the safety-bar condition.

## 2. Materials and Methods

### 2.1. Participants

Fourteen participants participated in this study, of whom 6 were men (26.3 ± 3.1 years, height 177.1 ± 6.9 cm, body mass: 81 ± 7.7 kg, fat percentage 16.11 ± 2.6%) and 8 were women (age 22.1 ± 2.2 years, height 166 ± 3.6 cm, body mass: 65.7 ± 10.5 kg, fat percentage 24.8 ± 4.3). Inclusion criteria were: (1) no injury or illness that could reduce maximal squat performance; (2) previously self-reported 1-RM at 1.5 (men) and 1.0 (women) times body mass in the last 12 months; (3) squat depth requirement meeting the International Powerlifting Federation Technical Rules book [9] guidelines, which requires the top surface of the thigh at the hip joint to be below the top of the knee joint when viewed laterally at the bottom of the squat; (4) performance of all three familiarization sessions and the test session; and (5) age 18–50 years. Furthermore, participants could not have performed any leg resistance training or consumed any alcohol 48 h before testing. The participants were informed orally and in writing of possible study risks. Written consent from each participant was obtained before the first familiarization session. The Regional Committee for Medical and Health Research Ethics deemed the study ethical without need for specific approval. Therefore, the study was performed according to the institutional ethical requirements, and approval for data security and handling was obtained from the Norwegian Center for Research Data (project: 701688) and in accordance with the latest revision of the Declaration of Helsinki.

### 2.2. Experimental Design

To investigate kinematic, kinetic, and myoelectric activity differences around the sticking region in the safety-bar, high-bar and low-bar squat conditions, a within-subjects, repeated measures design was used in which all participants performed all three conditions on one test day after several familiarization days. The dependent variables, barbell velocity, displacement, time, and joint angles, were collected as direct variables in the events v_0_, v_max1_, v_min_, and v_max2._ The dependent variables—myoelectric activity, moment arms, forces, and joint moments—were collected as means during the pre-sticking, sticking, and post-sticking region.

### 2.3. Protocol

A 3-RM test was used to investigate the different patterns around the sticking region in kinematics, kinetics, and myoelectric activity during a high-bar, low-bar, and safety-bar squat. Three familiarization sessions were conducted before the test session to ensure proper performance during test days and to find the actual 3-RM.

#### 2.3.1. Familiarization Days

On familiarization day 1, bi-acromion length was measured to decide the stance width for every participant. For this, 0.7 times of the bi-acromion length was used as stance width and standardized during all familiarization sessions and testing. External rotation of the foot was self-selected by the participants during the study but was then standardized and controlled for the rest of the study. The depth requirement was standardized through all familiarizations and test sessions using the rulebook of the International Powerlifting Federation [9]. Grip width was measured as the horizontal distance from the middle of the barbell to the first metacarpal. The barbell placement for the low-bar was measured as the distance in an axial direction from the C7 spinous process of the vertebra to the barbell. During familiarization sessions, participants were tested in the three squat variations on each familiarization session to find their individual 3-RM. A randomizer was used for the squat variations and 3-RM test order for the squat variations during the familiarization and test sessions (www.randomizer.org, accessed on 31 July 2021). During 3-RM familiarization sessions, both mean concentric barbell velocity and the repetitions in a reserve-based rating of perceived exertion [10] were measured and used for ensuring true 3-RMs were achieved. Mean concentric barbell velocity was calculated on the last repetition using a linear encoder. In particular, mean concentric barbell velocity of the final repetition during each 3-RM test was recorded to ensure related, maximum velocities in every testing condition. Both the familiarization and test days started with a general warm-up, which consisted of three sets of 6–10 repetitions with an Olympic barbell (Rogue, Ohio power bar). Throughout all three familiarization sessions, participants rested for 180 s during warm-up sets and 240 s between maximal lifting sets. Between the familiarization sessions, participants had a minimum of 96 h of rest and 120 h of rest between the test sessions. This was to prevent unnecessary exhaustion that could affect performance.

#### 2.3.2. Test Day

On test day, the participants performed a standardized warm-up protocol with the first randomized squat condition. The standardized squat protocol started with taking the lowest obtained familiarization 3-RM for calculating the following: four repetitions at 40%, three repetitions at 55%, and three repetitions at 70% of the lowest obtained familiarization 3-RM. Thereafter, load was increased by 1–10 kg for the randomized squat condition to achieve the recorded mean concentric barbell velocity or if failure occurred on the third repetition. After achieving a 3-RM in a squat condition, the participants tested the next squat condition starting at the lowest obtained familiarization 3-RM in the other squat conditions until true 3-RM were achieved. After all lifts were performed, participants completed a 5-s maximal isometric squat in the bottom position with the same stance widths as the dynamic trials for electromyography (EMG).

### 2.4. Measurements

A linear encoder (ET-Enc-02, Ergotest Technology AS, Langesund, Norway) was used to measure the lifting time of the barbell, and the vertical displacement was measured from the lowest point of the barbell with a resolution of 0.019 mm at a 200 Hz sampling rate. The velocity of the barbell was calculated by using the 5-point differential filter with software (MuscleLab version 10.200.90.5095, Ergotest innovation, Porsgrund Norway). The hip, knee, and ankle angles and moments, barbell displacement, and velocity were identified at the following positions during the ascent of the squat during the last repetition: lowest position of the barbell (v_0_), first maximal barbell velocity (v_max1_), first located lowest barbell velocity (v_min_), and second maximal barbell peak velocity (v_max2_). The linear encoder was synchronized with the EMG recordings using a MuscleLab 6000 system and analyzed by MuscleLab version 10.200.90.5095 (Ergotest Technology AS, Langesund, Norway).

#### 2.4.1. Electromyography (EMG) Measurements

EMG activity was recorded with MuscleLab 6000 (Ergotest Technology AS, Langesund, Norway). A preamplifier located as close to the pickup point as possible amplified and filtered the raw EMG signals. The signals were high-pass and low-pass (500, 20 Hz) filtered, rectified, and integrated. Maximal voluntary isometric contraction (MVIC) EMG activity was measured on the participants’ dominant side, placed with SENIAM recommendations [11] on the following muscles: erector spinae iliocostalis, erector spinae longissimus, gluteus maximus, gluteus medius, semitendinosus, biceps femoris, adductor longus, rectus femoris, vastus lateralis, vastus medialis, gastrocnemius medialis, and soleus medialis. Before placing the gel-coated self-adhesive electrodes (Dri-Stick Silver circular sEMG Electrodes AE-131, NeuroDyne Medical, Cambridge, MA, USA), the skin was shaved, rubbed, and washed with alcohol. The electrodes (11 mm contact diameter and 2 cm center-to-center distance) were placed along with the assumed orientation and direction of the underlying muscle fiber corresponding to the recommendations by SENIAM [11]. The raw EMG signals were converted to the root of mean square (RMS) signals with a hardware circuit network, which had a common rejection rate of 106 dB. Commercial software (MuscleLab V8.13, Ergotest Technology AS, Langesund, Norway) to analyze the stored EMG data. The mean RMS was calculated for the pre-sticking, sticking, and post-sticking region [12]. For normalization, the participants performed a 5 s MVIC high-bar squat in the bottom position at the same stance width, depth, and barbell placement. The barbell was attached to a squat rack, which could be corrected axially. Participants were told to achieve maximal force as fast as possible and maintain the maximal force through the test. The EMG activity was calculated as mean RMS between 2 and 4 s.

#### 2.4.2. Kinetics and Kinematic Measurements

A three-dimensional (3D) motion capture system (Qualysis, Gothenburg, Sweden), with eight cameras sampling at a frequency of 500 Hz, was used to track reflective markers, creating a 3D positional measurement. The 3D motion capture system was synchronized with the linear encoder and EMG recordings, using the MuscleLab 6000 system (Ergotest Technology AS, Langesund, Norway). Markers were placed on both sides of the body, except for the upper and lower hand which were placed on the lateral and medial epicondyle of the humerus and the radial and ulna styloid process. For the thorax, markers were placed on the C7 spinous process of the vertebra, acromion, TV1 thoracal process of the vertebra, the midpoint between the inferior angles of the most caudal points of the two scapulae, and sternum xiphisternal and sternum jugular notch joint [13]. For the upper and lower arm segment, markers were placed on the lateral and medial portion of the radial and ulna styloid process and epicondyle of the humerus. Markers for the pelvis were located on the posterior superior iliac spine and anterior superior iliac spine, creating a coda pelvis and hip joint center [14,15]. For the foot and shank, markers were placed on the lateral and medial malleolus, first and fifth proximal phalanx, and the femoral lateral and medial epicondyle. To track the events v_0_, v_max1_, v_min_, and v_max2_, four markers were placed on the barbell with a 20 cm distance. To track the 3D ground reaction forces and enable inverse dynamics calculation of joint moments, two force plates (AMTI Multi-axis Force Transducer BP6001200-2000, Lexington, KY, USA: Kistler force plate, type 9260AA6, Winterthur, Switzerland) were integrated in the Qualisys motion capture system. The origin of the axes was set to the corner of the left force platform. The x, y, and z axes were set to mediolateral, anterior-posterior, and vertical directions, respectively.

For segment modelling and analyses in Visual 3D v6 software (C-motion, Germantown, MD, USA), motion capture data were exported as C3D files. A low-pass Butterworth filter at a cut-off frequency of 10 Hz was used for smoothing all calculations from the model-based data. The torso, hip, knee, and ankle, angles in the different events (v_0_, v_max1_, v_min_, and v_max2_) were analyzed and calculated with a Cardan sequence in the order x-y-z, from distal to proximal orientation. Torso angle was calculated as the angle between the torso segment and the lab, and angles for the hip, knee, and ankle joints were calculated as the angle between the distal and proximal segments (Figure 1). Utilizing inverse dynamics calculations in a resolute coordinate system, the 3D joint moments for the hip, knee, and ankle were calculated. The joint moments were calculated as internal net joint moments with regard to the distal segments resolute coordinate system. The joint moments were reported as means and standard deviations at the different events to examine how the joint moments shifted at each region. The right and left segments sum was used to calculate net joint moments data. Net joint moments from the frontal plane are knee abduction and from the sagittal plane are flexion and extension moments. The analyzed plane net joint moments were normalized to the mass of participants using default normalization and expressed as Nm/kg. Moment arms were calculated as the anterioposterior distance between the hip, knee, and ankle joints and the ground reaction force vector.

### 2.5. Statistical Analysis

Normality was tested using the Shapiro–Wilk test. To assess differences between sexes in load lifted, an independent samples *t*-test was performed. To assess differences in barbell and joint kinematics, a 3 (conditions: safety-bar, low-bar, and high-bar) × 4 (events: v_0_, v_max1_, v_min_, and v_max2_) × 2 (sexes: men and women) analysis of variance (ANOVA) was used. To assess differences in EMG activity and joint moments between various conditions and regions, a 3 (conditions) × 3 (regions) ANOVA with repeated measures was performed. If significant differences were found, a Holm–Bonferroni post-hoc test was performed. In cases where the sphericity assumption was violated, a Greenhouse–Geisser adjustment for *p*-values was reported. The significance level was set at *p* ≤ 0.05. Statistical analysis was performed in SPSS version 21.0 (SPSS, Chicago, IL, USA). All results are presented as means ± standard deviations if not otherwise stated. The effect size was evaluated with *η*^2^ (Eta partial squared) where 0.01 < *η*^2^ < 0.06 constitutes a small effect, a medium effect when 0.06 < *η*^2^ < 0.14, and a large effect when *η*^2^ > 0.14 [16].

## 3. Results

The average 3-RM load lifted by participants was 91 ± 30.1 kg, 93.4 ± 9 kg, and 97.6 ± 30.5 kg with the safety-bar, high-bar, and low-bar squat conditions, respectively. The average 3-RM loads lifted for men were 116.1 ± 14.5 kg, 118.5 ± 12.7 kg and 123.3 ± 12.1 kg with the safety-bar, high-bar, and low-bar squat conditions. For women, the average 3-RM loads lifted were 71.8 ± 23.3 kg, 74.5 ± 22.4 kg, and 78.4 ± 25.3 kg with the safety-bar, high-bar, and low-bar squat conditions. A significant effect of load lifted was found between the different squat conditions and sexes (F ≥ 7.8, *p* ≤ 0.006, η^2^ ≥ 0.55). Men lifted significantly more with the high-bar, low-bar, and safety-bar condition compared with women (*p* ≤ 0.001). The participants lifted significantly more with the low-bar condition compared with the two other squat conditions (*p* ≤ 0.012). Also, the participants lifted significantly more load with the high-bar condition compared with the safety-bar (*p* ≤ 0.007). No significant differences were observed between the squat conditions or sexes in velocities, distances, or timing at the ascent phase of the events around the sticking region (F ≤ 3.4, *p* ≥ 0.059, η^2^ ≤ 0.19) (Figure 2).

A significant effect of torso and hip flexion angle was found during all events (F ≥ 5.9, *p* ≤ 0.023, η^2^ ≥ 0.31). The low-bar condition had the greatest torso angle followed by the high-bar and safety-bar conditions during all events (*p* ≤ 0.023) (Figure 3). For hip flexion angles, post-hoc tests showed that the participants performed less hip flexion for the safety-bar squat in v_0_, v_max1_, and v_max2_ compared to the other squat conditions (*p* < 0.021), while in v_min_, the low-bar condition had greater hip flexion than the other conditions (*p* ≤ 0.026). For knee extension angles, a significant effect was found for the events v_0_ and v_max2_ (F ≥ 4.5, *p* ≤ 0.02, η^2^ ≥ 0.24), while no significant effects occurred during v_max1_ and v_min_ (F ≤ 2.9, *p* ≥ 0.071, η^2^ ≤ 0.17). Bonferroni post-hoc tests revealed that the participants squatted with greater knee extension when using the safety-bar condition compared to the low-bar condition at v_0_ and v_max2_ (*p* ≤ 0.019). A significant effect of squat condition was found for ankle dorsi flexion angle (F ≥ 6.2, *p* ≤ 0.004, η^2^ ≥ 0.31). Bonferroni post-hoc tests revealed that the safety-bar condition had greater ankle dorsiflexion than the low-bar condition in v_max2_ (*p* = 0.004). No other significant difference was found in hip extension, knee extension and ankle dorsi flexion angle for all squat conditions (F ≤ 1.2, *p* ≥ 0.329, η^2^ ≤ 0.08). For knee valgus there was a significant sex difference found at v_max2_ in the high-bar and safety-bar conditions (F ≥ 1.93, *p* ≤ 0.04, η^2^ ≥ 0.31) where men had higher knee valgus than women (Figure 4). No other significant differences were found between men and women for any joint kinematic measurements for any squat conditions during any events (F ≤ 3.13, *p* ≥ 0.08, η^2^ ≤ 0.19).

No significant effect of squat condition was found for moment arms in the hip during the different events (F ≤ 1.66, *p* ≤ 0.21, η^2^ ≤ 0.11). A significant effect of squat condition was found for mean knee moment arms at v_0_, v_min_, and v_max2_ (F ≥ 4.52, *p* ≥ 0.02, η^2^ ≥ 0.24). Post-hoc tests showed that at v_min_ and v_max2_ the safety-bar condition had a larger moment arm compared to the low-bar condition (*p* ≤ 0.006). However, no significant differences were found at v_0_ and v_max1_ for knee moment arms (*p* ≥ 0.112). Furthermore, a significant difference at v_min_ was found, with a greater knee moment arm with the high-bar condition compared to safety-bar (*p* = 0.025). No further significant differences were found between the safety-bar and high-bar conditions for knee moment arms during the different events (*p* ≥ 0.08). A significant effect of squat condition was found for mean ankle moment arms at v_0_ and v_min_ (F ≤ 3.63, *p* ≥ 0.04, η^2^ ≤ 0.21). A post-hoc test showed significant differences in ankle moment arms between the high-bar and low-bar conditions at v_0_ and v_min_ (*p* ≤ 0.032), where the low-bar condition had a greater ankle moment arm than the high-bar.

A significant effect of events on the hip moment arm was also found (F ≥ 131.9, *p* ≤ 0.001, η^2^ ≥ 0.9). Bonferroni post-hoc tests revealed that the hip moment arm decreased significantly from v_min_ to v_max2_ for all squat conditions (*p* ≤ 0.001, Figure 5). When analyzing the knee moment arm during the different events, a significant effect of the event was found (F ≥ 90.85, *p* ≤ 0.001, η^2^ ≥ 0.86). Post-hoc tests revealed that the knee moment arm decreased from v_max1_ to v_min_ and v_max2_ for all squat conditions (*p* ≤ 0.001). However, no other significant interactions with events were found for mean hip, knee, or ankle moment arms for any squat conditions (F ≥ 0.74, *p* ≥ 0.53, η^2^ ≥ 0.05).

No significant effect of squat condition was found for mean or peak hip extension or ankle plantar flexion moments during the ascent (F ≤ 2.4, *p* ≥ 0.111, η^2^ ≤ 0.16). However, a significant effect of sticking region was found (F ≥ 15.4, *p* ≤ 0.001, η^2^ ≥ 0.252). Bonferroni post-hoc tests revealed that the hip extension moment decreased from region to region for all squat conditions (*p* ≤ 0.001), while the plantar flexion and knee extension moment decreased from the pre-sticking to the sticking and post-sticking regions (*p* ≤ 0.014) (Figure 6). Furthermore, a significant effect between conditions was found upon the mean descent and ascent knee extension moment (F ≥ 3.6, *p* ≤ 0.042, η^2^ ≥ 0.20), but not for the peak knee extension moment (F = 0.14, *p* = 0.78, η^2^ = 0.01). Post-hoc tests showed that the knee extension moment was greater when squatting with a safety-bar condition than with the low-bar condition during both the descent and ascent phases (*p* ≤ 0.042).

No significant effects were found for squat condition on myoelectric activity except for the gluteus maximus (F = 3.9, *p* = 0.039, η^2^ = 0.16). Squatting with a safety-bar led to greater myoelectric activity of the gluteus maximus than squatting with the high-bar technique (*p* = 0.024), with no differences between the other conditions (*p* ≥ 0.05). Furthermore, a significant effect of the region was found for myoelectric activity of the erector muscles, gluteus maximus, quadriceps, hamstring, and shank muscles (F ≥ 3.9, *p* ≤ 0.039, η^2^ ≥ 0.16). No significant differences were found in myoelectric activity between the regions for the gluteus medius or adductor longus muscles (F ≤ 1.9, *p* ≥ 0.166, η^2^ ≤ 0.76) (Figure 7). Bonferroni post-hoc tests revealed that both erector muscles increased myoelectric activity in the sticking region for all squat conditions (*p* ≤ 0.02). Furthermore, the gluteus maximus and biceps femoris myoelectric activity increased during each region (*p* ≤ 0.028), while the semitendinosus increased myoelectric activity from the pre-sticking to the post-sticking region for all three squat conditions. The quadriceps myoelectric activity decreased from the pre-sticking and sticking region to the post-sticking region (*p* ≤ 0.011), while the gastrocnemius and soleus decreased myoelectric activity from the pre-sticking to the post-sticking region (*p* ≤ 0.028) (Figure 8). A significant effect of sexe was found for the gluteus maximus during the safety-bar and high-bar conditions (F ≥ 6.1, *p* ≤ 0.008, η^2^ ≥ 0.36), where post-hoc tests revealed a higher gluteus maximus myoelectric activity in the safety-bar condition for men (pre-sticking: 83.7 ± 40.7%, sticking: 150.6 ± 65.4%, and post-sticking: 203.6 ± 103%) than women (pre-sticking: 53.5 ± 30.7%, sticking: 93.3 ± 37.7%, and post-sticking: 191.1 ± 43.4%) in all regions (*p* ≤ 0.04). For the high-bar condition, a significant difference in the post-sticking region was observed, where higher myoelectric activity in the gluteus maximus was observed for men (130.1 ± 19.8%) compared to women (102 ± 40.1) (*p* = 0.046). No other significant differences were found in myoelectric activity between the sexes for the other muscles in any region during any squat condition (F ≥ 1.9, *p* ≤ 0.166, η^2^ ≤ 0.09).

No significant effect of squat condition was found for mean force output at the different events (F ≤ 2.57, *p* ≥ 0.09, η^2^ ≤ 0.19). However, a significant interaction between events was found for force output between the different squat conditions (F ≤ 30.56, *p* ≥ 0.001, η^2^ ≤ 0.74), where post-hoc tests showed that force output decreased from v_0_ to all other events for all squat conditions (*p* ≤ 0.001). No other significant differences were found for force output in the other events for any squat condition (*p* ≥ 0.17).

## 4. Discussion

The aim of the study was to compare barbell and joint kinematics, kinetics, and myoelectric activity of the safety-bar condition with the high-bar and low-bar squat around the sticking region. The main findings were that the participants lifted the least load with the safety-bar followed by the high-bar and low-bar squat. Also, greater myoelectric activity of the gluteus maximus was observed during the safety-bar squat compared to the high-bar squat. In addition, differences were found in joint kinematics for the knee extension moments between the safety-bar and the low-bar squat. Finally, men had higher myoelectric activity in the safety-bar condition for the gluteus maximus during all sticking point regions in comparison with women, and more knee valgus than for women at v_max2_.

No significant differences were observed between the squat conditions in barbell kinematics at the ascent phase or the events around the sticking region, and the findings for barbell kinematics were similar to previous studies investigating the sticking region in back squats [8,12,17,18,19,20]. Performing the safety-bar squat allowed the most upright torso position followed by the high-bar and then the low-bar squat, which is in agreement with Hecker, Carlson and Lawrence [2]. The more upright torso angle may be ideal for decreasing the shear forces in the lower back [21], when performing a safety-bar squat. The participants performed the safety-bar squat with less hip flexion at v_0_, v_max1_, and v_max2_ compared to the other squat conditions, while at v_min_, the low-bar condition had greater hip flexion than the other squat conditions. Previous studies comparing high-bar and low-bar squats reported similar results, where the low-bar condition had a greater torso angle compared to the high-bar condition [22,23,24]. Since the low-bar placement reduces the hip moment arm, it results in a greater torso inclination to produce the same moment around the hip joint (Figure 3). This causes a more anterior displacement of the center of pressure, which is visible in the increased moment arm of the ankle (since all squats were maximal lifts, the hip moments were maximal). The center of pressure is the total pressure acting on a body and causes a force that acts through that point. A small difference between the high-bar and safety-bar conditions is caused by moving the arms forward, which would theoretically cause an increased moment arm. However, this does not occur likely due to the combination of a lower load lifted during safety-bar squats and/or a lower moment arm due to a more upright position (Figure 5). This change of angles to establish the same hip moment causes a more anterior projection of the center of pressure, which is apparent when viewing the longer moment arm in the ankle joint in the low-bar condition compared with the other two squat conditions. This anterior projection produces a lower moment arm on the knee joint, which increases during the safety-bar condition (Figure 5).

The gluteus maximus was the only muscle that showed significant myoelectric differences between the high-bar and safety-bar conditions. This likely occurred due to a more upright position in the safety-bar condition. This upright position during the safety-bar condition may have resulted in more ideal joint angles putting the gluteus maximus in a better position to improve its force-length properties (Figure 7). As previously discussed, this is also thought to be one of the reasons for the sticking region’s existence. Since the gluteus maximus could not be used early in the pre-sticking region due to the force-length relationship, the quadriceps may be emphasized more at the start of the lift. Consequently, the safety-bar condition produces more myoelectric activity for the gluteus maximus with less load lifted. This is an interesting finding, which is in contrast with the finding by Hecker, Carlson and Lawrence [2], where lower myoelectric gluteus maximus activity was found for the safety-bar squat compared to a traditional barbell squat. However, Hecker, Carlson and Lawrence [2] performed both conditions at equal absolute loads, which may have altered the EMG comparison.

The myoelectric activity of erector muscles increased in the sticking region for all squat conditions, while myoelectric activity increased during each region for all squat conditions for the gluteus maximus and biceps femoris (Figure 7). The quadriceps’ myoelectric activity decreased from the pre-sticking and sticking region to the post-sticking region, while the gastrocnemius and soleus increased myoelectric activity from the pre-sticking to the post-sticking region for all three squat conditions (Figure 8). The findings are in accordance with earlier studies [8,12,17,18,19,20,25]; however, the bi-articular rectus femoris muscle, which is a hip flexor and knee extensor, had high myoelectric activity in the pre-sticking region for all squat conditions, which is dissimilar to the results from Robertson, et al. [26]. They observed that the rectus femoris’ myoelectric activity did not change during the entire ascent phase of the lift. However, the present study compared 3-RM squats, while Robertson, Wilson and Pierre [26] analyzed squats at 80% of 1-RM, which may alter the demands on different muscles. Furthermore, we observed increases in biceps femoris myoelectric activity during the pre-sticking to the post-region for all squat conditions (Figure 8), possibly contributing to the sticking region due to this co-contraction of the knee extensors and flexors. No differences were found for adductor longus or gluteus medius myoelectric activity between conditions or regions (Figure 7 and Figure 8). This may have occurred because the rotation angle for the foot was controlled between the squat conditions after initial participant self-selection. Therefore, it is possible that adductor longus myoelectric activity is impacted more by the hip rotation angles than barbell placement.

When investigating the joint moments during the different regions of the sticking phase, the largest hip, knee, and ankle moments were found at v_0_ for all three squat conditions (Figure 6), which is in line with the findings by Robertson, Wilson and Pierre [26]. The hip moment decreased from the pre-sticking to the post-sticking region for all conditions (Figure 6), which was also similar to the results of Robertson, Wilson and Pierre [26]. Peak knee and ankle moments were also the highest at the bottom position; however, and then decrease through the subsequent regions in all three squat conditions. Since the peak hip, knee, and ankle moments are all at the bottom position, it could be thought that this would be the hardest position since it is the weakest point to exert force. However, peak knee moments occur at v_0_ and v_max1_ during the ascent phase, which may not impact squat performance to a considerable degree since the transition from eccentric to concentric at the bottom position of the squat is aided by elastic/stretch reflex contributions to force production. This is demonstrated by greater acceleration occurring during the transition to concentric following the eccentric. Afterwards, force remains similar throughout the ascent due to similar accelerations at all subsequent events (force = mass **×** acceleration) (Figure 9). The relationships between moments and myoelectric activity around the sticking region provides further insight regarding the muscular contributions during these phases.

Peak hip moments occurred at v_0_ and v_max1_, while the myoelectric activity of the gluteus maximus was lower in the pre-sticking region than in the sticking region and the hip moment decreased in the post-sticking region (Figure 6). It seems that the gluteus maximus is an agonist in the sticking region, indicated by increased myoelectric activity from the pre-sticking to the sticking region. However, quadriceps myoelectric activity decreased from the pre-sticking and sticking region to the post-sticking region for all squat conditions, indicating that during the first part of the ascent, the quadriceps are the prime movers. The ascent phase of the lift starts with the quadriceps muscles and soleus performing knee extension and plantar flexion, which decreases around the lowest velocity time point (v_min_), where the gluteus muscles and hamstrings increases their myoelectric activity. The reason for the gluteus muscles’ low activity at the start of the ascending phase is most likely due to the large muscle length and moment arms, causing a mechanically poor position to exert force (Figure 9). Also, our findings showed that gluteus maximus myoelectric activity first peaked in the post-sticking region, indicating that the muscle’s capability to exert force and contribute to the hip extensor moment may not be optimal during the sticking region even if the hip extensor moment demands is large in this region. This coincides with the findings from van den Tillaar, Kristiansen and Larsen [7], who found that the gluteus maximus peaked in myoelectric activity first at 0.25 m barbell height in 1-RM Smith machine squats, which is where the post-sticking region began for all squat conditions in our study. Therefore, our results show that the combination of large hip moments and a disadvantageous force production position for the gluteus maximus may be a notable contributor to the sticking region, independent of barbell placement.

There were two interesting differences found between the sexes. First, there was a difference in myoelectric activity for the gluteus maximus between sexes, in which the gluteus maximus in the safety-bar condition for all regions was higher in activity for men and can be an important variable for the difference in load lifted. Second, there was a higher knee valgus for men compared to women, which can come from differences in certain movement characteristics between the sexes. Between men and women there are some differences in hip anatomy that are worth mentioning, which may help explain differences in knee valgus. For instance, the female pelvis is generally shorter and wider than the male pelvis, which is taller and narrower [27,28]. Also, men tend to have less hip range of motion compared to women, which can lead men to have movement compensations when working through a full range of motion [29,30]. Due to anatomical and neuromuscular factors, greater knee valgus (knee moving inwards) for women than for men is often observed during squatting movements [31]. Women often have greater lumbar lordosis angles at 7–13 degrees and standing sacral slope [32,33]. This places men’s and women’s glutes in different positions, and suggests that the female lumbar spine (on average) has more range of motion. While we observed a greater valgus in men compared to women, seemingly in contrast with previous research, it is possible that the observed difference may have come from the standardized stance of 0.7 times horizontal acromion length. Since men have longer distances between their acromions compared to women, this could lead to greater relative stance widths. With smaller hips, a harder time moving through the range of motion, and a greater absolute stance width, this may have caused more knee valgus among men. Also, the observation of greater knee valgus with increasing stance width is supported by Lahti, et al. [34], who found greater knee valgus with wide stance compared to narrow stance. This is a limitation of the study and therefore, future studies should use hip width to determine the stance width when comparing kinematics and kinetics between the sexes.

## 5. Conclusions

Use of a safety-bar resulted in a more upright torso inclination compared to the high-bar squat, possibly resulting in a more advantageous position for the gluteus maximus to contribute to the hip extensor moment. Moreover, squatting with a low-bar technique allowed for more loads to be lifted, followed by the high-bar and safety-bar squat. This finding may be attributed to the larger hip moments and similar knee moments compared to the other squat conditions. Therefore, we recommend squatting with a low-bar technique when the goal is to lift the most load possible among recreationally trained men and women. Also, our results indicate that when targeting myoelectric activity in the gluteus maximus, squatting with a safety bar may be more effective than the high-bar condition.

## Figures and Tables

**Figure 1 ijerph-18-08351-f001:**
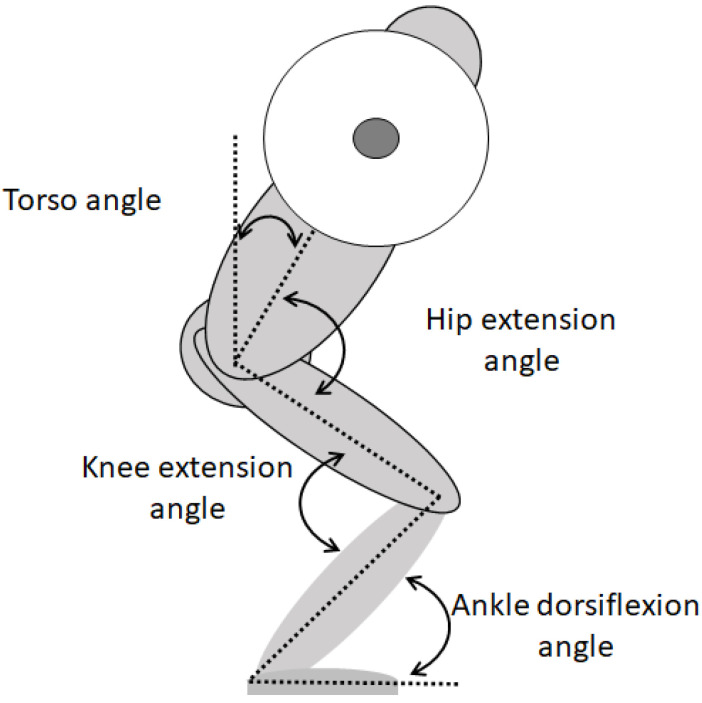
Different joint angles during a back squat.

**Figure 2 ijerph-18-08351-f002:**
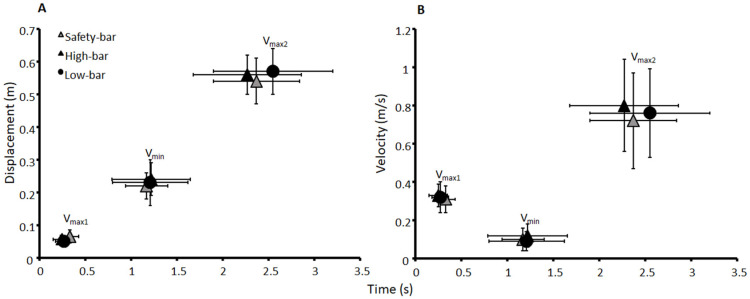
Mean ± standard deviation (SD) (**A**) displacement and (**B**) velocity at the events v_max1_, v_min_, and v_max2_, and their timings for the high-bar, low-bar, and safety-bar squat during 3-RM.

**Figure 3 ijerph-18-08351-f003:**
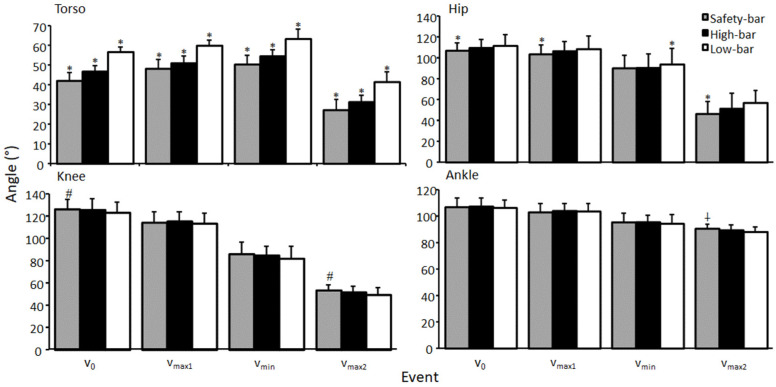
Mean (±SD) torso, hip extension, knee extension, and ankle flexion angle at the events v_max1_, v_min_, and v_max2_ for the high-bar, low-bar, and safety-bar squat during 3-RM. * indicates a significant difference between this squat condition and all other squat conditions on a *p* ≤ 0.05 level. # indicates a significant difference between this squat condition and the low-bar condition on a *p* ≤ 0.05 level. † indicates a significant difference between this squat condition and low-bar condition on a *p* ≤ 0.05 level.

**Figure 4 ijerph-18-08351-f004:**
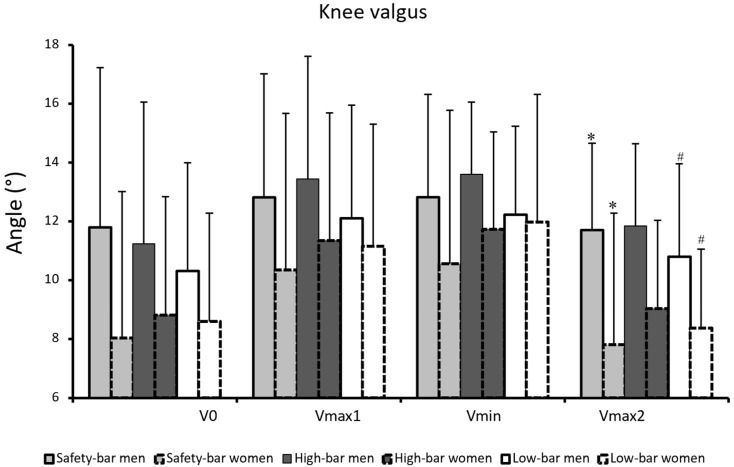
Mean (±SD) knee valgus at the events v_max1_, v_min_, and v_max2_ for the high-bar, low-bar, and safety-bar squat during 3-RM. * indicates a significant difference between sexes in safety-bar squat condition on a *p* ≤ 0.05 level. # indicates a significant difference between sexes in high-bar squat condition on a *p* ≤ 0.05 level.

**Figure 5 ijerph-18-08351-f005:**
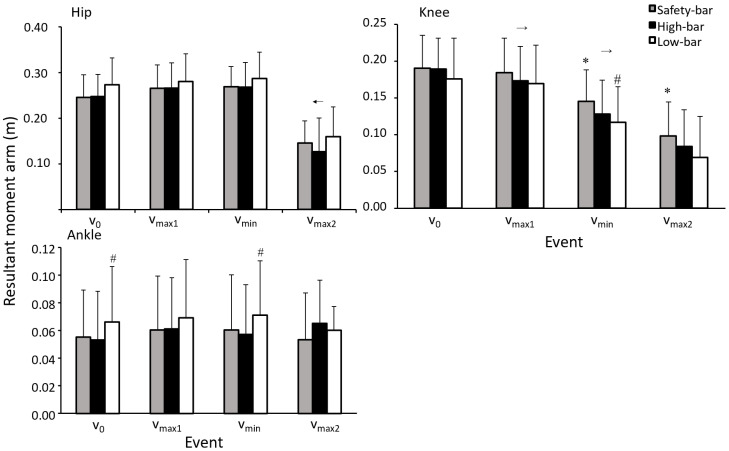
Mean (±SD) hip, knee, and ankle moment arms at the events v_max1_, v_min_, and v_max2_ for the high-bar, low-bar, and safety-bar squat during 3-RM. * indicates a significant difference between this squat condition and low-bar squat on a on a *p* ≤ 0.05 level. # indicates a significant difference between this squat condition and the high-bar condition on a *p* ≤ 0.05 level. → or ← a significant difference between this event and all the other events for all squat conditions on a *p* ≤ 0.05 level.

**Figure 6 ijerph-18-08351-f006:**
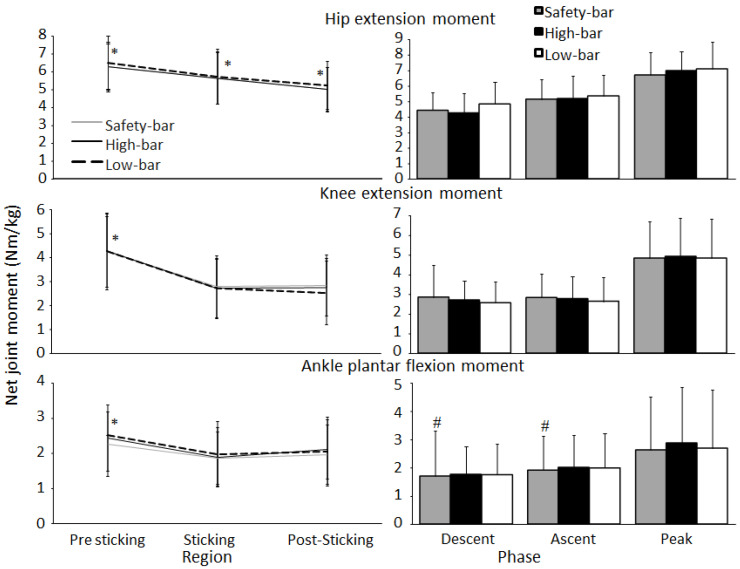
Mean ± SD net joint moments during descent, ascent, pre-sticking, sticking, post-sticking region together with peak net joint moments for the high-bar, low-bar, and safety-bar squat during 3-RM. * indicates a significant difference between this region and all other regions on a *p* ≤ 0.05 level for all squat conditions. # indicates a significant difference between this squat condition and the low-bar condition on a *p* ≤ 0.05 level.

**Figure 7 ijerph-18-08351-f007:**
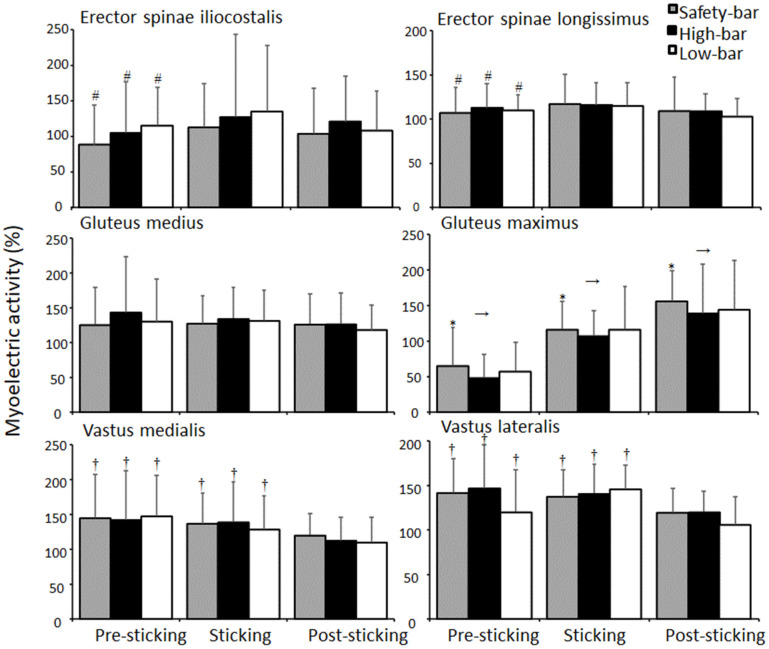
Mean ± SD normalised myoelectric activity for erector spinae iliocostalis, erector spinae longissimus, gluteus medius, gluteus maximus, vastus medialis and vastus lateralis during pre-sticking, sticking, and post-sticking region for the high-bar, low-bar, and safety-bar squat during 3-RM. # indicates a significant difference between the pre-sticking and sticking region for all squat conditions on a *p* ≤ 0.05 level. * indicates a significant difference between this squat condition and the high-bar condition on a *p* ≤ 0.05 level. → indicates a significant difference between this region and all other regions for all squat conditions on a *p* ≤ 0.05 level. † indicates a significant difference between this region and the post-sticking region for all squat conditions on a *p* ≤ 0.05 level.

**Figure 8 ijerph-18-08351-f008:**
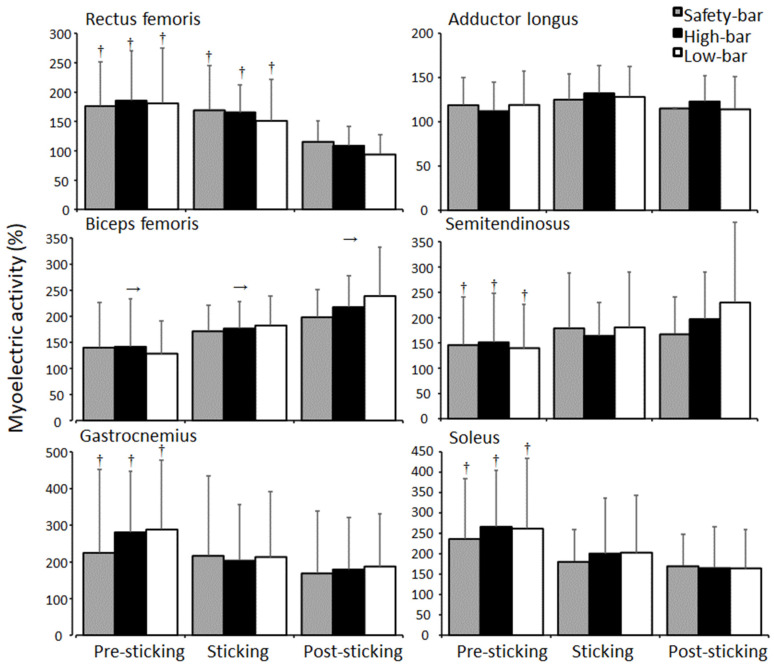
Mean ± SD normalised myoelectric activity for rectus femoris, adductor longus, biceps femoris, semitendinosus, gastrocnemius, and soleus during pre-sticking, sticking, and post-sticking region for the high-bar, low-bar, and safety-bar squat during 3-RM. → indicates a significant difference between this region and all other regions for all squat conditions on a *p* ≤ 0.05 level. † indicates a significant difference between this region and the post-sticking region for all squat conditions on a *p* ≤ 0.05 level.

**Figure 9 ijerph-18-08351-f009:**
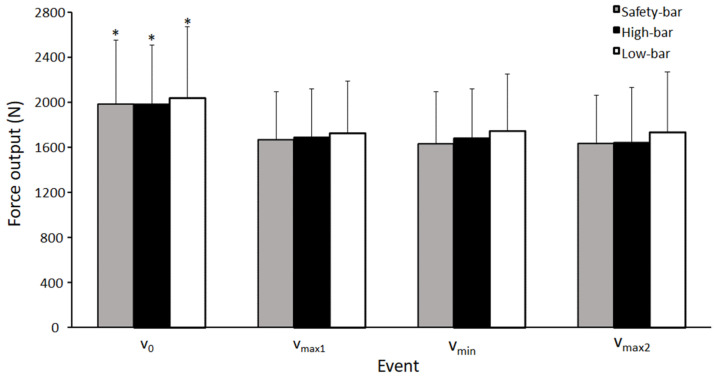
Mean (±SD) force output at the events v_0,_ v_max1_, v_min_, and v_max2_ for the high-bar, low-bar, and safety-bar squat during 3-RM. * indicates a significant difference between this event and all other events for all squat conditions on a *p* ≤ 0.05 level.

## Data Availability

The data presented in this study are available on request from the corresponding author. The data are not publicly available due to rules of Norwegian Center for Research Data.

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
