# Peer review of "A Biomechanical Comparison of the Safety-Bar, High-Bar and Low-Bar Squat around the Sticking Region among Recreationally Resistance-Trained Men and Women"

_ijerph, 2021, doi:10.3390/ijerph18168351_

Round 1
Reviewer 1 Report
The study is interesting and to some extent it is a continuation of the previous work done by the authors. It looks carefully planned and performed; the data obtained were processed with rigour. The results are discussed and compared to other studies on similar topic. Experts focused on sports performance and training might find the conclusions important for their practice.
Some specific comments and suggestions might be helpful to improve the way, how the work is presented to the reader:
L85-87: Cohen 1992 (A power primer) presented some guidelines regarding the sample size that are perhaps worth of future considerations.
L105: The symbols for the events analysed are not clear at this point. They are introduced in L151-154. Also, it would be beneficial, if an exemplary diagram of velocity with events noted is included in Results section, as well as other quantities measured (EMG).
L146: Was there any particular reason why the linear encoder was used, if you have motion capture data from Qualisys?
Perhaps Figure 1 (it could be a table as well) could me moved between L195-196. It doesn’t seem related to 2.2 paragraph.
L195: … and vertical directions, respectively.
L204: How were the mass properties of the body segments considered and how were the moment arms estimated (a figure in Methods section could be included)?
2.4 Statistical analysis: This section seems rather unclear to the reader. Could you please state more clearly, what the factors and what the dependent variables were in each case? What test exactly was used to assess the differences between sexes assessed?
Figure 2: Are A and B panels switched?
In Figure 7 caption, you introduce a left and a right arrow – only the latter is used in the figure …
L393: Are you sure it's the right figure referenced?
Figure 10: It would be nice to annotate the differences, which are not that obvious at the first glance. It would make the figure more meaningful.
L404: …, Since …; L407: This an interesting … Spell check might be beneficial.
L486-488: Probably, the hip width could be estimated from the motion capture data to avoid the limitation?
Author Response
We would like to thank the reviewer for reviewing our paper with detailed feedback. We have now made several changes, and we believe that the paper now is suited for publication in your journal.
Reviewer 1
The study is interesting and to some extent it is a continuation of the previous work done by the authors. It looks carefully planned and performed; the data obtained were processed with rigour. The results are discussed and compared to other studies on similar topic. Experts focused on sports performance and training might find the conclusions important for their practice.
Some specific comments and suggestions might be helpful to improve the way, how the work is presented to the reader:
L85-87: Cohen 1992 (A power primer) presented some guidelines regarding the sample size that are perhaps worth of future considerations.
Thank you. We did not calculate power, but we will have that in mind in future studies.
L105: The symbols for the events analysed are not clear at this point. They are introduced in L151-154. Also, it would be beneficial, if an exemplary diagram of velocity with events noted is included in Results section, as well as other quantities measured (EMG).
We have now included description about the events and the pre-sticking, sticking, and post-sticking region in the introduction. We already have a figure about the velocity development in the results and think that this is enough information, to avoid repetition from previous studies and figures on these events (figure 2)
L146: Was there any particular reason why the linear encoder was used, if you have motion capture data from Qualisys?
It was used for getting instant mean concentric velocity after a maximal lift attempt, for ensuring true 3-RM lift. Also, it was used for synchronizing data since EMG recordings was done in another program (Musclelab)
Perhaps Figure 1 (it could be a table as well) could me moved between L195-196. It doesn’t seem related to 2.2 paragraph.
We have deleted figure 1 since it does not give so much extra information then described in the text. We hope the reviewer agrees about this.
L195: … and vertical directions, respectively.
This is now corrected.
L204: How were the mass properties of the body segments considered and how were the moment arms estimated (a figure in Methods section could be included)?
We have now included a sentence about the moment arms in the methods “Moment arms from were calculated as the anterioposterior distance between the hip, knee, and ankle joints and the ground reaction force vector. “
The mass properties is automatically calculated in v3d with assumptions from previous studies. We only added the bodymass on the barbell and the participants when modelling, which is normal procedure. We cannot define our own mass properties because we don’t have the equipment for doing such analyzes.
2.4 Statistical analysis: This section seems rather unclear to the reader. Could you please state more clearly, what the factors and what the dependent variables were in each case? What test exactly was used to assess the differences between sexes assessed?
This section is now updated more cleary, and also we have included the test used to test sex differences.
Figure 2: Are A and B panels switched?
Yes, we have now corrected this.
In Figure 7 caption, you introduce a left and a right arrow – only the latter is used in the figure …
Thank you for notifying, this has now been corrected.
L393: Are you sure it's the right figure referenced?
This is now corrected.
Figure 10: It would be nice to annotate the differences, which are not that obvious at the first glance. It would make the figure more meaningful.
The data for figure 10 was already given in figure 5. Thus, we refer to this figure instead in the text and deleted figure 10 to avoid confusion.
L404: …, Since …; L407: This an interesting … Spell check might be beneficial.
This is now corrected to “this is an interesting”.
L486-488: Probably, the hip width could be estimated from the motion capture data to avoid the limitation?
Absolute. As written, this is a limitation of the study since we based the stance width upon the acromion width. This we can do in future studies.
Reviewer 2 Report
This paper presents a biomechanical study of 25 subjects performing three kinds of squats with safety-bar, high-bar, and low-bar conditions. The work is exciting and well-presented, but I concern about the methodology and the results, as follows:
The variables V0, Vmax1, Vmin, and Vmax2 are not adequately defined, and also, they should justify why they are relevant in this context.
Figure 1 should be a table, and the images do not represent squats or the conditions here evaluated. The experimental design is baffled. The authors should elaborate better on the experimental design and justify referencing state of the art.
Figure 1 and other figures are not mentioned or adequately discussed in the text.
Figures 1 (On page 6) and 10 are feeble. Therefore, I suggest representing the exercise here performed with authentic images from the experiments clarifying all the conditions and defining all the variables related.
On page 5, section 2.4, the paper presents an ANOVA. It should perform a normality test firstly. This section should be better elaborated.
Author Response
This paper presents a biomechanical study of 25 subjects performing three kinds of squats with safety-bar, high-bar, and low-bar conditions. The work is exciting and well-presented, but I concern about the methodology and the results, as follows:
The variables V0, Vmax1, Vmin, and Vmax2 are not adequately defined, and also, they should justify why they are relevant in this context.
This is now clearly defined in the introduction together with the pre-sticking, sticking, and post-sticking region.
Figure 1 should be a table, and the images do not represent squats or the conditions here evaluated. The experimental design is baffled. The authors should elaborate better on the experimental design and justify referencing state of the art.
The first figure has been taken away since it had no extra meaning when most of it was clearly written in the text.
Figure 1 and other figures are not mentioned or adequately discussed in the text.
We have now mentioned all figures in the manuscript.
Figures 1 (On page 6) and 10 are feeble. Therefore, I suggest representing the exercise here performed with authentic images from the experiments clarifying all the conditions and defining all the variables related.
We decided to delete figure 10 since it represents only an average of the findings on the moment arms which are displayed already in the resultant moment arm figure 5. Thereby this figure would be double, and you never can have authentic images that represent an average participant. Figure 1 is only used to define the different angles used and does not need to be authentic, since its purpose is to define the different angles.
On page 5, section 2.4, the paper presents an ANOVA. It should perform a normality test firstly. This section should be better elaborated.
A normality test was performed, and this is now presented in the article. Also, the statistics has a better elaboration of the different testing performed.
Reviewer 3 Report
Summary
This study aimed to compare the kinematics, kinetics, and EMG activity during a back squat when using a safety bar or high bar/low bar position. Twenty-five participants took part in the study and performed all bar type and position type squatting movements. This study found that greater loads could be lifted with the low bar position and least with the safety bar. There was greater glut max activity when using the safety bar. There were some differences between men and women for load lifted, EMG activity of the glut max, and knee valgus movement.
Study Importance
A previous study compared the safety bar to back squat but did not discern whether the high bar or low bar position was used. Therefore, this comparison is missing.
General Comments
Recommend reordering the introduction information so that the portion concerning the safety bar (line 35-49) comes before the purpose. This would provide a better lead in to the purpose of the study and why the safety bar is being included.
Was the safety bar weight accounted for in the total load lifted? Was the weight of the safety bar the same as the traditional bar? In my experience the safety bar can be heavier than traditional bar
The methods section is a bit confusing. It is difficult to discern protocols and measurements. For example, what was the protocol used to determine a 1RM, an actual 1RM test or estimated from a 3RM? Later in the methods it says a squat protocol was done using a % of the lowest 3RM measure. Was this a warm-up or the squat protocol for the measuring of kinematics, kinetics, and EMG? Recommend using subheading to improve clarity and keep working consistent when describing protocols. The methods need clear flow in the writing.
The experimental design would be more of an overview of what was done to answer the question purposed by the study. Recommend adding a protocol section for all the protocols performed with subheading for familiarization and experimental protocol and then the measurements section is only how all the measures were performed with subheading for kinematics, kinetics, and EMG.
There are two figure 1’s (page 3 and page 6) and only 1 is referenced in the manuscript. Further, not all figures are referenced in the text.
Some of the results are oddly reported. For example, knee flexion angle events seems to have multiple ANOVA statistics reported but shouldn’t this be 1 ANOVA test? An ANOVA table for the comparison of knee angle events, then post-hoc test results for each comparison. Ensure this is the case for all analyses. Another example, only post-hoc results for EMG activity at glut max was reported for safety bar vs. high bar, but not reported for safety bar vs. low bar or low bar vs high bar. Everything needs to be reported for post-hoc when the ANOVA table is significant even if post-hoc test lacks significance between some or all comparisons.
Recommend using tables rather than figures for reporting the results. Currently the figures make it look as if there is very little difference between conditions as the bars look like they end is similar spots across conditions. With tables, the p-values for within condition and/or event could be reported in their own columns. Partial etas could remain in the text with F-statistics with their p-values removed. Post-hoc results could be shown in the table using symbols and exact reporting in the text.
Were there any differences in training status between this study compared to other studies? Is it possible the degree of co-contraction mentioned in the discussion is more pronounced in the recreationally trained participants within this study compared to other studies that potentially used more well trained participants?
Recommend reviewing Murawa et al. 2020 Muscle activation varies between high-bar and low-bar back squat, Glassbrook et al. 2019 The high-bar and low-bar back squats: A biomechanical analysis, & Goodin et al 2017 Comparison of power and velocity in the high bar and low bar back squat across a spectrum of loads, to add to the manuscript.
Specific Comments
Line 74: I do not believe this statement is completely true and recommend re-wording. Though Hecker et al. did not specify the bar position, in saying back squat most would consider this to mean traditional bar placement (high bar placement), in my experience.
Line 88-89: previous reported 1-RM of 1.5 and 1.0x body weight were self-reported or from another study? If from another study, please provide the reference. If self-reported, please state so.
Line 105: V0, Vmax1, Vmin, and Vmax2 are difficult to read in their small text format. Recommend not subscripting the entire abbreviation only a portion. I.e. V0, Vmax1, Vmin, and Vmax2. These abbreviations also need defining before their use. What events are these in reference to during the squatting movement?
Figure 1: This figure makes it seem like this was a training study, but this study seems more acute. Also, the bench press icon is misleading as this was a squatting study. Recommend fixing to more accurate reflect the 3 familiarization periods performed before the measured trial. Was it one familiarization session before a measured trial or were 3 familiarization sessions before each measured trial? Ensure this is clear throughout the manuscript.
Line 110-115: Did the 3RM estimate the 1RM performed in Week 1 shown in the figure? If not, how was the 1RM measured? I do not see methods on this.
Line 122-123: What distance from the C7 meant the bar was in the low bar position? Was it a certain distance for everyone or a relative distance (%)? What constitutes a low bar position?
Line 127: What was used to measure barbell velocity? Need to equipment mentioned here.
Line 141-145: All squat conditions were performed on the same day? That is what it seems to say here. Thought this was done over 3 weeks like figure 1 shows? After all lifts were performed, then EMG was measured? So, all squat conditions were completed on the same day and then when very fatigued, EMG measures were determined? Reword if not true. This lacks clarity and would impact repeatability.
Line 181: specifically, where were markers placed on the dominate hand?
Line 227: missing units on first measure
Line 228-299: where is the data reported on what men lifted vs. women? At no point is the data broken down between males and females using mean/SD.
Line 231: what were the individual post-hoc p-values for the comparisons between low bar, high bar and safety bar? Ensure if 3 or more comparisons were found significant by the ANOVA that the post-hoc results were reported as well throughout the results section
Line 388: I do not believe COP was previously defined in the manuscript and since it is only used twice. It should be written out rather than abbreviated.
Line 403-404: Are these two separate sentences as the word “Since” is capitalized?
Line 411: EMG activity increased for erector muscles during what condition, all? It is specified in the later half for the other two muscle
Line 465-488: Currently, no results (mean ± SD with the associated p-values) are reported in the results section to support conclusion made in this paragraph. The degree of the differences are unknown
Line 465-470: recommend talking about one finding first then the other rather than going back and forth and saying first this, second that; first this, second that; first this, second that. Needs to be more concise.
Author Response
Reviewer 3
Summary
This study aimed to compare the kinematics, kinetics, and EMG activity during a back squat when using a safety bar or high bar/low bar position. Twenty-five participants took part in the study and performed all bar type and position type squatting movements. This study found that greater loads could be lifted with the low bar position and least with the safety bar. There was greater glut max activity when using the safety bar. There were some differences between men and women for load lifted, EMG activity of the glut max, and knee valgus movement.
Study Importance
A previous study compared the safety bar to back squat but did not discern whether the high bar or low bar position was used. Therefore, this comparison is missing.
General Comments
Recommend reordering the introduction information so that the portion concerning the safety bar (line 35-49) comes before the purpose. This would provide a better lead in to the purpose of the study and why the safety bar is being included.
We like the order of the introduction as it is. It gives a good lead in to the purpose of the study, and why the safety-bar is being included.
Was the safety bar weight accounted for in the total load lifted? Was the weight of the safety bar the same as the traditional bar? In my experience the safety bar can be heavier than traditional bar
Yes, if you refer to the mass of the safety-bar it was accounted for in the total load lifted. Also, load was not matched but intensity (3-RM).
The methods section is a bit confusing. It is difficult to discern protocols and measurements. For example, what was the protocol used to determine a 1RM, an actual 1RM test or estimated from a 3RM? Later in the methods it says a squat protocol was done using a % of the lowest 3RM measure. Was this a warm-up or the squat protocol for the measuring of kinematics, kinetics, and EMG? Recommend using subheading to improve clarity and keep working consistent when describing protocols. The methods need clear flow in the writing.
We performed 3 familiarization sessions were we tested 3-RM in the different squat conditions. In the test days we performed 3-RM, and not 1-RM or estimated 1-RM. Under we have pasted from the methods chapter. We don’t really understand how this is confusing?
“Three familiarization sessions were conducted before the test session to ensure proper performance during test days and to find the actual 3-RM.”
“During familiarization sessions, participants were tested in the three squat variations on each familiarization session to find their individual 3-RM.”
“During 3-RM familiarization sessions, both mean concentric barbell velocity and the repetitions in a reserve-based rating of perceived exertion [12] were measured and used for ensuring true 3-RMs were achieved. Mean concentric barbell velocity was calculated on the last repetition using a linear encoder. In particular, it was recorded mean concentric barbell velocity of the final repetition during each 3-RM test to ensure related, maximum velocities in every testing condition. Both the familiarization and test days started with a general warm-up, which consisted of three sets of 6–10 repetitions with an Olympic barbell (Rogue, Ohio power bar). Throughout all three familiarization sessions, participants rested for 180 seconds during warm-up sets and 240 seconds between maximal lifting sets. Between the familiarization sessions, participants had a minimum of 96 hours of rest and 120 hours of rest between the test sessions. This was to prevent unnecessary exhaustion that could affect performance.
On test days, the participants performed a standardized warm-up protocol with the first randomized squat condition. The standardized squat protocol started with taking the lowest obtained familiarization 3-RM for calculating the following: four repetitions at 40%, three repetitions at 55%, and three repetitions at 70% of the lowest obtained familiarization 3-RM. Thereafter, weight was increased by 1–10 kg for the randomized squat condition to achieve the recorded mean concentric barbell velocity or if failure occurred on the third repetition. After achieving a 3-RM in a squat condition, the participants tested the next squat condition starting at the lowest obtained familiarization 3-RM in the other squat conditions to true 3-RM was achieved. After all lifts were performed, participants completed a 5-second maximal isometric squat in the bottom position with the same stance widths as the dynamic trials for electromyography (EMG). “
The experimental design would be more of an overview of what was done to answer the question purposed by the study. Recommend adding a protocol section for all the protocols performed with subheading for familiarization and experimental protocol and then the measurements section is only how all the measures were performed with subheading for kinematics, kinetics, and EMG.
We have now added protocol and measurements as subheadings.
There are two figure 1’s (page 3 and page 6) and only 1 is referenced in the manuscript. Further, not all figures are referenced in the text.
The first figure has been deleted. The different figures have now been referred to in the text.
Some of the results are oddly reported. For example, knee flexion angle events seems to have multiple ANOVA statistics reported but shouldn’t this be 1 ANOVA test? An ANOVA table for the comparison of knee angle events, then post-hoc test results for each comparison. Ensure this is the case for all analyses. Another example, only post-hoc results for EMG activity at glut max was reported for safety bar vs. high bar, but not reported for safety bar vs. low bar or low bar vs high bar. Everything needs to be reported for post-hoc when the ANOVA table is significant even if post-hoc test lacks significance between some or all comparisons.
You are absolutely right. We have rewritten the statistics chapter. Moreover, we have also presented all post-hoc comparisons for the significant ANOVA results.
Recommend using tables rather than figures for reporting the results. Currently the figures make it look as if there is very little difference between conditions as the bars look like they end is similar spots across conditions. With tables, the p-values for within condition and/or event could be reported in their own columns. Partial etas could remain in the text with F-statistics with their p-values removed. Post-hoc results could be shown in the table using symbols and exact reporting in the text.
We respect your opinion on this. However, all the authors have discussed this. And we prefer figures, because it is easier to visualize the findings. Also, differences are not that large in most cases, which as you point out, our figures show.
Were there any differences in training status between this study compared to other studies? Is it possible the degree of co-contraction mentioned in the discussion is more pronounced in the recreationally trained participants within this study compared to other studies that potentially used more well trained participants?
The participants in this study were trained, however it can be a factor for the results where more well trained/elite participants can alter the results. But the participants were above the inclusion criteria and therefore se no problem in this study. Also, to ensure that the participants performed the different squat conditions properly, 3 familiarizations sessions were executed, which in most other studies just have 1 familiarization performed. Moreover, we don’t have the assumptions based on our data to speculate how the use of different cohorts could impact co-contraction.
Recommend reviewing Murawa et al. 2020 Muscle activation varies between high-bar and low-bar back squat, Glassbrook et al. 2019 The high-bar and low-bar back squats: A biomechanical analysis, & Goodin et al 2017 Comparison of power and velocity in the high bar and low bar back squat across a spectrum of loads, to add to the manuscript.
We have added the article by “Glassbrook et al. 2019 The high-bar and low-bar back squats: A biomechanical analysis” to our paper.
Specific Comments
Line 74: I do not believe this statement is completely true and recommend re-wording. Though Hecker et al. did not specify the bar position, in saying back squat most would consider this to mean traditional bar placement (high bar placement), in my experience.
Sorry, but here we disagree. We believe this depends on the cohort.
Line 88-89: previous reported 1-RM of 1.5 and 1.0x body weight were self-reported or from another study? If from another study, please provide the reference. If self-reported, please state so.
This was self-reported at the day of testing and not from another study. we have corrected it in the article.
Line 105: V0, Vmax1, Vmin, and Vmax2 are difficult to read in their small text format. Recommend not subscripting the entire abbreviation only a portion. I.e. V0, Vmax1, Vmin, and Vmax2. These abbreviations also need defining before their use. What events are these in reference to during the squatting movement?
We have defined the events earlier in the text now (introduction). These abbreviations with small text format are used in previous literature about this topic, and therefore is used in this study as well.
Figure 1: This figure makes it seem like this was a training study, but this study seems more acute. Also, the bench press icon is misleading as this was a squatting study. Recommend fixing to more accurate reflect the 3 familiarization periods performed before the measured trial. Was it one familiarization session before a measured trial or were 3 familiarization sessions before each measured trial? Ensure this is clear throughout the manuscript.
We have excluded the first figure since it was confusing for all reviewers and we think the information was already written in the text.
Line 110-115: Did the 3RM estimate the 1RM performed in Week 1 shown in the figure? If not, how was the 1RM measured? I do not see methods on this.
It was a 3-RM tested in this study not a 1-RM, 1-RM was just to determine that the strength level of the participants above the inclusion criteria. This was self-reported as mentioned in the text now.
Line 122-123: What distance from the C7 meant the bar was in the low bar position? Was it a certain distance for everyone or a relative distance (%)? What constitutes a low bar position?
A low-bar position means that the barbell is in a lower vertical barbell position on the back than in the high-bar position. This was measured in familiarization 1, and standardized thorough all familiarization and tests, as mentioned in the methods.
Line 127: What was used to measure barbell velocity? Need to equipment mentioned here.
We measured mean concentric velocity using a linear encoder. This is added in text under measurments
“A linear encoder (ET-Enc-02, Ergotest Technology AS, Langesund, Norway) was used to measure the lifting time of the barbell, and the vertical displacement was measured from the lowest point of the barbell with a resolution of 0.019 mm at a 200 Hz sampling rate. The velocity of the barbell was calculated by using the 5-point differential filter with software (MuscleLab version 10.200.90.5095, Ergotest innovation, Porsgrund Norway)”.
Line 141-145: All squat conditions were performed on the same day? That is what it seems to say here. Thought this was done over 3 weeks like figure 1 shows? After all lifts were performed, then EMG was measured? So, all squat conditions were completed on the same day and then when very fatigued, EMG measures were determined? Reword if not true. This lacks clarity and would impact repeatability.
Yes we tested in the same day. To ensure reliable data from EMG measurments and joint moments?
This is detailed described in the methods – protocols section:
“On test days, the participants performed a standardized warm-up protocol with the first randomized squat condition. The standardized squat protocol started with taking the lowest obtained familiarization 3-RM for calculating the following: four repetitions at 40%, three repetitions at 55%, and three repetitions at 70% of the lowest obtained familiarization 3-RM. Thereafter, load was increased by 1–10 kg for the randomized squat condition to achieve the recorded mean concentric barbell velocity or if failure occurred on the third repetition. After achieving a 3-RM in a squat condition, the participants tested the next squat condition starting at the lowest obtained familiarization 3-RM in the other squat conditions to true 3-RM was achieved. After all lifts were performed, participants completed a 5-second maximal isometric squat in the bottom position with the same stance widths as the dynamic trials for electromyography (EMG). “
Line 181: specifically, where were markers placed on the dominate hand?
This is now added in the text. “Markers for the upper and lower arm segment were placed on the lateral and medial epicondyle of the humerus and the radial and ulna styloid process”.
Line 227: missing units on first measure
Fixed.
Line 228-299: where is the data reported on what men lifted vs. women? At no point is the data broken down between males and females using mean/SD.
We have now added data on what men and women lifted in the different squat conditions.
Line 231: what were the individual post-hoc p-values for the comparisons between low bar, high bar and safety bar? Ensure if 3 or more comparisons were found significant by the ANOVA that the post-hoc results were reported as well throughout the results section.
P-values for individual post hoc-values are now added.
Line 388: I do not believe COP was previously defined in the manuscript and since it is only used twice. It should be written out rather than abbreviated.
COP is changed to centre of pressure in the entire manuscript now.
Line 403-404: Are these two separate sentences as the word “Since” is capitalized?
Yes. A mistake. This is changed now.
Line 411: EMG activity increased for erector muscles during what condition, all? It is specified in the later half for the other two muscles.
We have added “for all squat conditions.
Line 465-488: Currently, no results (mean ± SD with the associated p-values) are reported in the results section to support conclusion made in this paragraph. The degree of the differences are unknown
We have changed the figure of the valgus angles so it is easier to observe the significant differences at vmax2 for the high and safety conditions between men and women. We have also added the means and SD values with the associated p-values for the results section about the gluteus maximus.
“A significant effect of sex was found for the gluteus maximus during the safety-bar and high-bar conditions (F ≥ 6.1, p ≤ 0.008, η2 ≥ 0.36), whereas post-hoc tests revealed a higher gluteus maximus myoelectric activity in the safety-bar condition for men (pre-sticking: 83.7±40.7 %, sticking: 150.6±65.4 %, and post-sticking: 203.6±103 %) than women (pre-sticking: 53.5±30.7 %, sticking: 93.3±37.7 %, and post-sticking: 191.1±43.4 %) in all regions (p ≤ 0.04). For the high-bar condition, a significant difference in the post-sticking region was observed, where higher myoelectric activity in the gluteus maximus were observed for men (130.1±19.8 %) compared to women (102±40.1) (p = 0.046). No other significant differences were found in myoelectric activity between the sexes for the other muscles in any region during any squat condition (F ≥ 1.9, p ≤ 0.166, η2 ≤ 0.09).”
Also, we have also changed the valgus figure, so it is easier to interpret for the readers.
Line 465-470: recommend talking about one finding first then the other rather than going back and forth and saying first this, second that; first this, second that; first this, second that. Needs to be more concise.
We agree with the reviewer and re-arranged this part of the discussion.
Round 2
Reviewer 2 Report
This new version has clarified my concerns about the work. I recommend this paper for publication.
Reviewer 3 Report
No additional comments